# Using Patient-Specific 3D-Printed C1–C2 Interfacet Spacers for the Treatment of Type 1 Basilar Invagination: A Clinical Case Report

**DOI:** 10.3390/biomimetics10060408

**Published:** 2025-06-17

**Authors:** Tim T. Bui, Alexander T. Yahanda, Karan Joseph, Miguel Ruiz-Cardozo, Bernardo A. de Monaco, Alexander Perdomo-Pantoja, Joshua P. Koleske, Sean D. McEvoy, Camilo A. Molina

**Affiliations:** Department of Neurosurgery, Washington University School of Medicine, 660 S Euclid Ave, Campus Box 8057, St. Louis, MO 63110, USA; btim@wustl.edu (T.T.B.); ayahanda@wustl.edu (A.T.Y.); jkaran@wustl.edu (K.J.); miguelr@wustl.edu (M.R.-C.); abernardo@wustl.edu (B.A.d.M.); alexander.p@wustl.edu (A.P.-P.); joshua.koleske@wustl.edu (J.P.K.); smcevoy@wustl.edu (S.D.M.)

**Keywords:** atlantoaxial instability, basilar invagination, cervical spine, craniovertebral junction, custom implant, pediatric spine, three-dimensional printing

## Abstract

**Background:** Type 1 basilar invagination (BI) is caused by a structural instability at the craniovertebral junction (CVJ) and has been historically treated with distraction and stabilization through fusion of the C1–C2 vertebrae. Recent advances in 3D printed custom implants (3DPIs) have improved the array of available options for reaching distraction and alignment goals. **Case Presentation:** We report the case of a 15-year-old male who presented with early signs of cervical myelopathy. Radiographic evaluation revealed type 1 BI with a widened atlantodental interval (ADI) of 3.7 mm and a 9 mm McRae’s line violation (MLV) of the dens, resulting in severe narrowing at the CVJ and brainstem/spinal cord impingement. Of note, the patient had bilateral dysplastic C1 and C2 anatomy, thus requiring a patient-specific 3DPI to conform to this anatomy and enable sufficient distraction and fusion. Custom 3D printed C1–C2 interfacet spacers were created and implemented within 14 days to achieve sufficient distraction, osteoconduction, and stabilization of the C1–C2 joint. **Outcome:** Postoperatively, the patient remained neurologically intact with myelopathic symptom improvement before discharge on postoperative day 4. Postoperative imaging demonstrated the resolution of BI from successful C1–C2 joint distraction and confirmed intended implant placement with resolution of canal stenosis. During his 6-week follow-up, the patient remained neurologically stable with intact hardware and preserved alignment. **Conclusions:** This case is the first in the United States demonstrating the use of custom 3D printed interfacet spacers to achieve successful distraction, decompression, and stabilization of type 1 BI. These patient-specific 3DPIs were designed and created in a streamlined manner and serve as proof-of-concept of pragmatic implant design and manufacturing. Future optimization of the workflow and characterization of long-term patient outcomes should be explored for these types of 3DPI.

## 1. Introduction

Basilar invagination (BI) is a craniovertebral junction (CVJ) pathology in which the odontoid process abnormally projects through the foramen magnum. The resultant crowding of the brainstem may yield symptoms such as headache, dysphagia, limited neck motion, or sequelae of cervical myelopathy. BI is the most common CVJ malformation, comprising approximately 50% of such cases [1].

Importantly, BI can be clinically divided into either type 1 or type 2 based on the underlying etiology of odontoid prolapse [2]. Type 1 BI is a result of C1–C2 instability that leads to cranial migration of the dens into the foramen magnum. In contrast, type 2 BI is a consequence of platybasia and flattening of the skull base, clival hypoplasia, cranial settling, or other acquired or congenital conditions that affect skull base bone quality [3]. Type 2 BI is also associated with conditions such as Chiari-1 malformation. These distinctions between the types of BI are important when considering surgical management in symptomatic cases, as each type is treated differently [4,5,6].

Since type 1 BI is caused by atlantoaxial instability, treatment typically consists of traction to attempt to reduce the dens more caudally, followed by C1–C2 fusion (occipital–cervical fusion has also been described in the literature), potentially with spacers between the C1–C2 joints to provide additional distraction [4,7]. Type 2 BI may be treated with posterior fossa decompression with duraplasty (with/without coagulation of the cerebellar tonsils if there is a Chiari malformation), with limited benefit from preoperative traction [2,4]. In extreme cases of cranial settling, cervical or occipital–cervical fusion may be required [4,8].

Three-dimensional (3D) printing in spine surgery has evolved over time. Early applications were limited to preoperative planning or teaching with 3D models [9]. More recently, improvements in technology have enabled the creation of patient-specific 3D-printed implants (3DPIs) [10,11]. These implants, since they are bespoke to a patient’s needs, can integrate more seamlessly into patient anatomy and improve surgical workflow [10]. While a precedent has been established for use of custom 3DPIs in spinal oncology or deformity surgery, we were only able to identify one prior study from China evaluating custom 3DPIs for basilar invagination that featured persistent risk of subsidence [12]. Thus, we report the case of a pediatric patient with type 1 BI who was treated with C1–C2 decompression and fusion utilizing bilateral custom 3D-printed interfacet spacers to distract the dens. We furthermore discuss the rationale and benefit of using this novel technology in the treatment of BI and compare our implant design to the previously described Chinese implant.

## 2. Patient Information and Presentation

### 2.1. Clinical Presentation

A 15-year-old male with a history of behavioral disorders, bicuspid aortic valve, and genitourinary abnormalities presented to a neurosurgery clinic for anomalies noted on cervical spine radiographs. These images were obtained during a previous chiropractic visit that the patient’s family sought for potential management of his ADHD and ODD symptoms. The patient denied neck pain, paresthesia, extremity weakness, or gait disturbances. On physical examination, he was neurologically intact except for bilateral hyperreflexia in both the upper and lower extremities, which was concerning for early onset cervical myelopathy.

### 2.2. Imaging and Diagnosis

Cervical spine computed tomography (CT) revealed widening of the atlantodental interval (ADI) and basilar invagination (BI). The dens was displaced superiorly and posteriorly. The degree of BI was measured to be 9 mm superior to McRae’s line, or 9 mm of McRae Line Violation (MLV). ADI was measured at 3.7 mm. His clivo-axial angle (CXA) was 119° (Figure 1). There was severe narrowing at the CVJ with impingement upon the lower brainstem and upper cervical spinal cord.

Of note, the patient was noted to have abnormal osseous anatomy of the skull base and cervical spine, including a malformed clivus, hypoplastic posterior C1 arch, hypoplastic inferior C1 lateral masses, dysplastic superior articulating facets of C2, and incomplete segmentation of the C5 and C6 vertebrae (Figure 2). CT angiogram demonstrated a left-dominant vertebral artery with a diminutive right V4 segment and aberrant right subclavian artery.

Head and cervical spine magnetic resonance imaging (MRI) verified the CT findings, demonstrating compression of the upper cervical cord and cervicomedullary junction with associated T2/STIR hyperintensity indicative of spinal cord damage (Figure 3). There was no Chiari malformation or syringomyelia. Dynamic MRI demonstrated that neck flexion increased the ADI to 6 mm as the AP diameter of the C1 vertebral foramen decreased to 3 mm. Neck extension reduced the ADI to 3 mm with C1 diameter increasing to 6 mm. Despite clival hypoplasia and decreased CXA, both of which can be associated with type 2 BI, the atlanto-axial instability and rostral dens migration yielded a pathophysiology most consistent with type 1 BI. Nonetheless, it is important to note the continuous spectrum along which these disease processes present, as they may not always fit strictly within a strict category.

## 3. Materials and Methods

### 3.1. Preoperative Planning and 3DPI Design

Even though the patient was subjectively neurologically asymptomatic, the severe canal stenosis, MRI signal change, headaches with neck pain, and objective myelopathy on exam all pointed towards cervical myelopathy and atlanto-axial instability. Additionally, it was determined that posterior fossa decompression would not relieve the primary ventral compression and would not restore stability to the C1–C2 joint. Accordingly, a posterior C1–C2 fusion with C1 laminectomy was planned with the goal of also distracting the dens away from the skull base to diminish the BI and relieve cervicomedullary junction compression while restoring stability. The patient’s dysplastic anatomy within the C1–C2 joint, along with the degree of distraction of the dens that would be required, presented the opportunity to develop custom 3D-printed bilateral titanium interface spacers for precise execution of the surgical goals (Figure 4).

Photon-counting detector CT (PCD-CT) scans were performed using a Siemens NAEOTOM Alpha system (software version VB10A; Siemens Healthineers, Germany). Cervical spine (C-spine) studies were reconstructed at two slice thicknesses—3 mm and 0.4 mm—using the Br44f (medium-sharp) and Br64f (ultra-sharp) kernels, respectively. All reconstructions incorporated Quantum Iterative Reconstruction (QIR) at level 3. Additional details may be found in the Appendix A. These scans were used to create Digital Imaging and Communications in Medicine (DICOM) data. The DICOM was assessed with Radiology Workstation by the 3D Printing Center at Washington University in St. Louis. A computer-aided design (CAD) was created after accounting for anatomic defects and implementing surgeon input with Materialise Mimics 25.0 and SolidWorks 2020 software (Figure 4). A custom 3D model was created and underwent simulated finite element analysis of predicted load, fatigue, and other anticipated biomechanical forces. The analysis included extreme scenarios of minimal support from the patient’s remaining bone, in which the implant and supporting screws must endure the entirety of the force.

A propriety industry protocol from PURI-TI (DeGen Medical, Charleston, SC, USA) was followed to manufacture, ship, and sterilize this 3DPI more rapidly than conventionally manufactured 3DPIs. The implant was printed additively (avoiding cutting oils or material transfer), synthesized with titanium, and designed for incorporation with fusion adjuncts (e.g., bone graft, osteo-inductive/osteo-conductive materials). Implants underwent additional load and deformation testing before use in surgery. Because of its proprietary nature, the nuances of PURI-TI 3DPI manufacturing cannot be further discussed in this paper. The material composition of the implant was derived from biocompatible titanium and tested per ISO 10993 to ensure biocompatibility of both the material and manufacturing process. The methods and materials were previously accepted and used with other biocompatible implants. The 3D-printed devices were steam sterilized and underwent successful validation testing for 1-time-use devices per industry standard.

Institutional Review Board approval for compassionate use at the authors’ institution and FDA compassionate use approval was acquired within the 14 days between the time of device conception to use during surgery. The steps for obtaining initial FDA approval have been reported by our group in prior publications [11].

### 3.2. Surgical Approach

The patient was positioned prone with Gardner–Wells tongs applying 10 lbs of vertical traction to the skull. Preoperative neuromonitoring signals demonstrated decreased MEP/SSEP from the right upper extremity, but these signals did not diminish upon placement of traction or prone positioning. Fluoroscopy was used to show that no undue traction was caused after weights were placed.

A posterior C1–C2 subperiosteal exposure was performed, and a laminar spreader was used to augment the distraction between the skull and C2 vertebral body. The dorsal epidural space was accessed after resection of the ligamentum flavum in the C1–C2 intervertebral space. The bilateral C2 nerve roots were ligated proximal to the dorsal root ganglion and transected, and the remaining C2 nerve root sleeves were retracted medially to access the bilateral C1–C2 joint complex. A C1 laminectomy was performed for decompression of the cervicomedullary junction. The C1–2 complex bony borders were meticulously defined to ensure safe and appropriate manipulation of the joint and subsequent insertion of the implant. The C1–C2 joints were decorticated using serially larger rasps, which also prepared the facet joints for insertion of the custom spacers. A 7.5 mm height custom 3D-printed spacer was inserted between each C1–C2 joint to successfully accomplish distraction of the dens. These cages were packed with autograft and bone morphogenetic protein 2 (BMP-2). Freehand bilateral C1 lateral mass screws and C2 pedicle screws were placed, followed by the insertion of titanium rods spanning C1 to C2. Relevant posterior lateral gutter surfaces were decorticated and grafted with autograft, allograft, and BMP-2.

## 4. Results

The patient tolerated the procedure well and remained neurologically intact with resolution of myelopathic signs before discharge on postoperative day four. Postoperative imaging at the time of hospital discharge demonstrated placement of the implants as designed with resolution of pathologic BI and restoration of canal patency (Figure 5). Postoperative ADI was 3 mm, MLV was 2 mm, and CXA was 134°. In other words, the ADI decreased by 0.7 mm, MLV decreased by 7 mm, and CXA increased by 15° (Figure 6). A sagittal and coronal 3D render of the postoperative CT clearly illustrated successful implant insertion with sufficient distraction inserted between the C1/C2 joint. The intricate situation of the 3DPI and bony anatomy may be visualized in the C1/C2 articulating space (Figure 7).

At his 6-week follow-up appointment, the patient was doing well without postoperative pain. His incision was healing well. He had not experienced recurrence of myelopathic symptoms. Standing radiographs (Figure 8) showed intact hardware, proper positioning of implants, and preserved spinal alignment.

## 5. Discussion

Complex spinal pathology may necessitate atypical approaches or surgical techniques to address unique anatomical or other intricacies [13,14,15]. In the past, surgical options were limited by existing technology. More recently, 3DPIs have helped address this problem, enabling surgeons to create replacement vertebral bodies, interbody cages, sacropelvic implants, and more [16]. Even more novel is the ability to produce custom, patient-specific 3DPIs, which can conform precisely to anatomic defects that are either defined preoperatively or created intraoperatively [17]. In this report, we describe the use of custom 3DPI to address the need for odontoid distraction via interfacet spacers for the treatment of BI. This the first report of this technology in the United States.

Several prior publications highlight either the use of 3DPIs in general or patient-specific cases for larger surgeries including large fusions, corpectomies, en bloc tumor resections, or sacrectomies [10,16]. These studies have shown that 3DPIs may routinely be used to address osseous defects, provide appropriate fusion rates, and achieve desired postoperative outcomes. In general, 3DPIs are more expensive than conventional instrumentation, and their creation may require turnaround times of several weeks. Our institution has shown that it is possible to achieve an expedited design and manufacturing process [10]. Overall, for complicated cases associated with longer operative times, greater surgical invasiveness, or greater post-operative complications, 3DPIs are a useful surgical option when applicable [10,16,18,19]. Regarding cervical spine or CVJ, 3DPIs are less common, with relatively sparse literature on their applications.

Type 1 BI may prove difficult to treat given multiple potential pathologies at the CVJ, cervicomedullary junction, and cervical spine. Surgical approaches may be somewhat limited by a narrow operative corridor. The earliest iterations of surgical management of type 1 BI involved cervical decompression alone. However, this often is insufficient, as exhibited by our case where cervical decompression would not fully address the impingement of the dens upon the cervicomedullary junction and underlying mechanical instability. Interfacet spacers provided adequate distraction that obviated the need for more morbid and poorly tolerated surgeries such as a transoral approach for resection of the dens [20].

The first interfacet spacers for CVJ pathology, introduced by Goel, were limited in their height and ability to achieve proper cervical alignment [21]. More modern designs, while eventually reaching alignment goals, were limited by additional intraoperative time and labor requirements via intraoperative adjustment of the device or anatomical sites [1]. These issues are compounded by patient-specific anatomic anomalies—such as in this case, the hypoplastic and dysplastic doming of the C1–C2 joint. Type 1 BI, then, is a condition that could benefit immensely from custom 3DPIs.

3DPIs have been used in multiple ways to address atlantoaxial instability [12,22]. Niu et al. described a case utilizing a 3D-printed craniovertebral fixation device for occipital–cervical fusion for upper cervical deformity and atlantoaxial dislocation [22]. Additionally, Jian et al. published a series of 14 patients in China who received 3D-printed C1–C2 intra-articular cages during fusions for various cases of atlantoaxial instability, many of whom had BI [12]. Jian et al. evaluated bullet-type cages, which were designed in a wedged “bullet” shape with smooth surfaces for facile joint insertion (Figure 9). However, bullet-type cages have a relatively small footprint, potentially increasing risks of subsidence via increased force along a focal contact-point [23]. Our current case employed technology similar to that used by Jian but with a broader base to minimize subsidence risks (Figure 4). Due to our patient’s young age and the fact that his BI was due to instability specifically around the C1–C2 junction, we elected to perform a C1–C2 fusion rather than the more morbid occipital–cervical fusion, thereby significantly sparing cervical range of motion.

Subaxial cervical spine interfacet spacers as a treatment for radiculopathy or as a fusion adjunct have been reported previously [24]. However, far less common are C1–C2 interfacet spacers for the additional goal of distraction of the dens as a treatment for BI. In our case, a patient-specific 3DPI was required over a more conventional interfacet spacer for multiple reasons. The patient had hypoplastic and domed superior articulating facets bilaterally, producing an abnormally small and rounded surface area for articulation with C1, which required custom spacer design for a proper fit. Specifically, design goals of the implant included sufficient implant height to achieve the distraction goal, rounded surfaces to conform to joint anatomy, a maximized graft window without compromising implant mechanical integrity, maximizing implant footprint for load sharing, and optimized insertional workflow, which included angled implant insertors to navigate the abnormally oblique craniocaudal angulation of the joint (Figure 4).

Further investigation is required to discern the nuances in outcomes for custom 3D-printed C1–C2 interfacet spacers in the treatment of BI or related pathology. This solution has not been previously reported outside of China, and this serves as a feasibility report demonstrating the technical and regulatory capability of performing patient-specific reconstructions in the axial cervical spine. A larger series will be needed to elucidate further relevant outcomes such as implant subsidence and fusion rates.

## 6. Conclusions

We present a case report of an adolescent patient who received patient-specific 3D-printed interfacet spacers for treatment of type 1 BI. The 3DPIs achieved proper alignment and sufficient decompression in postoperative imaging. The design, manufacturing, approval, and implantation process occurred over 14 days, illustrating the feasibility of precise and streamlined implementation. This case was the first of its kind in the United States to demonstrate effective decompression and alignment of BI with a customized implant in a patient with severe BI and cord compression. Further exploration of 3DPI for BI with a deeper dive into optimization of the design/manufacturing process and characterization of long-term patient outcomes should be explored as this technology becomes increasingly used in clinical practice.

## Figures and Tables

**Figure 1 biomimetics-10-00408-f001:**
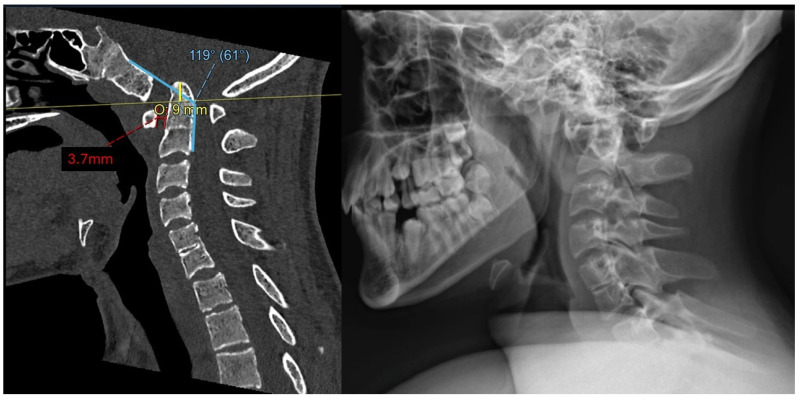
Preoperative mid-sagittal cervical spine CT (**left**) and X-ray (**right**) demonstrate superior and posterior translation of the dens consistent with BI (9 mm McRae’s line violation, 3.7 mm ADI, and CXA of 119°).

**Figure 2 biomimetics-10-00408-f002:**
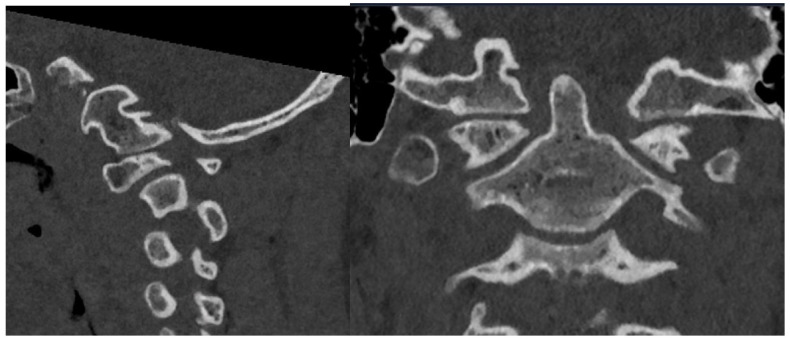
Preoperative sagittal (**left**) and coronal (**right**) CT demonstrating irregular anatomy of cervical spine, including malformed clivus, hypoplastic posterior C1 arch, hypoplastic inferior C1 lateral masses, and dysplastic C2 superior articulating facets.

**Figure 3 biomimetics-10-00408-f003:**
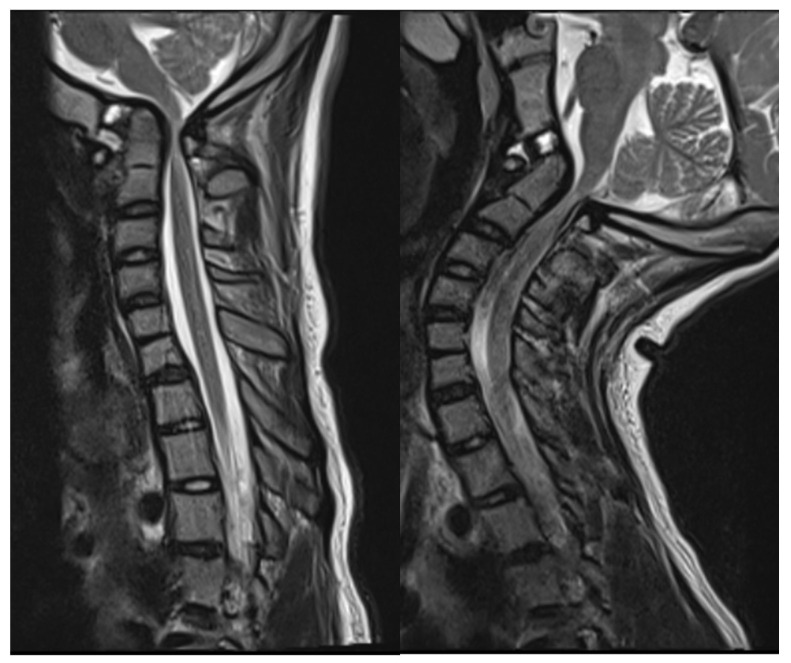
Preoperative mid-sagittal T2 MRI of caudal brain and cervical spine in neutral (**left**) and extension (**right**) demonstrating severe canal stenosis and T2/STIR hyperintensity of the upper cervical cord.

**Figure 4 biomimetics-10-00408-f004:**
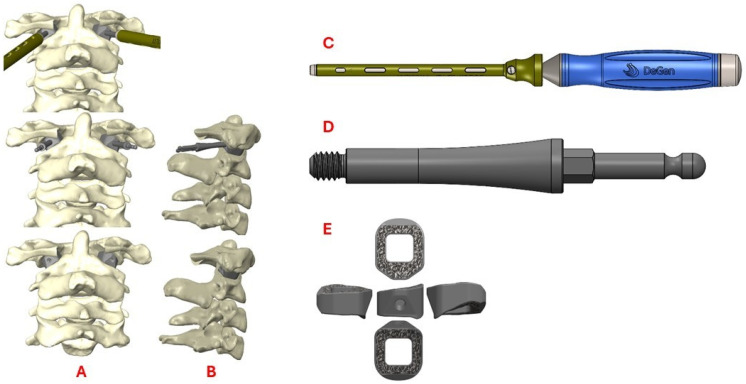
Schematics demonstrating coronal (**A**) and sagittal (**B**) views of the custom 3D-printed titanium implants at various stages of insertion using a fixation pin driver (**C**) and insertion adapter (**D**). A 3D rendering of the implant (**E**) is also shown.

**Figure 5 biomimetics-10-00408-f005:**
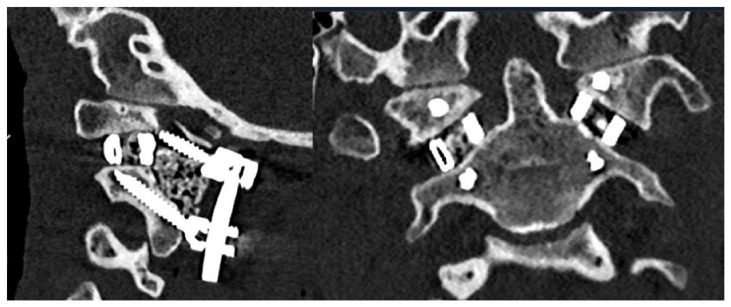
Sagittal (**left**) and axial (**right**) postoperative CT demonstrating successful implant insertion with corresponding rods and screws.

**Figure 6 biomimetics-10-00408-f006:**
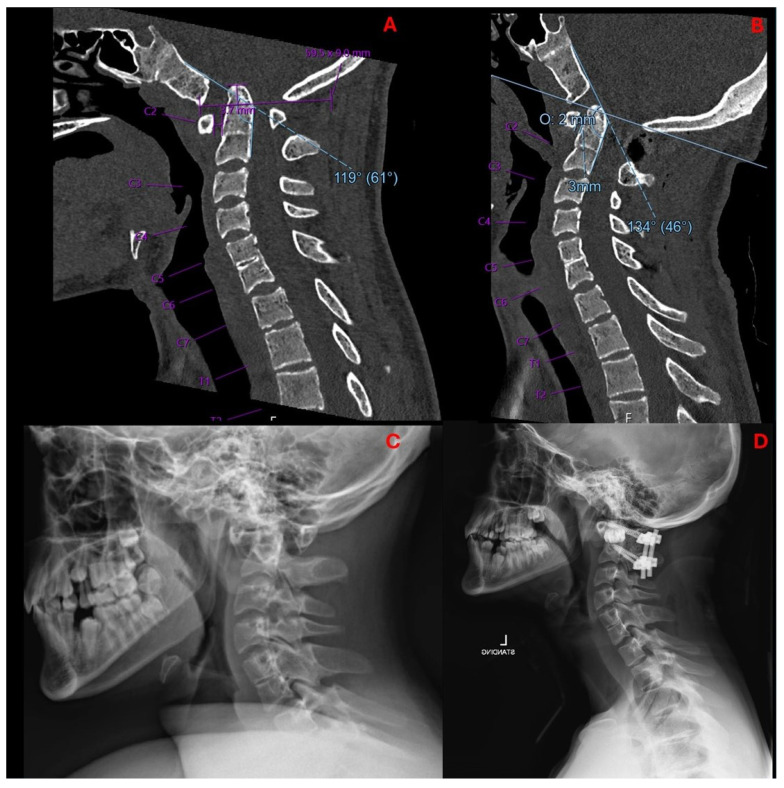
Mid-sagittal cervical CT pre- (**A**) and postoperatively (**B**) demonstrate decreased basilar invagination (preop 5 mm, postop 2 mm), decreased ADI (preop 3.7 mm, postop 3 mm), and increased CXA (preop 119°, postop 134°). Lateral radiographs pre- (**C**) and postoperatively (**D**) demonstrated proper positioning and decompression from 3DPI insertion.

**Figure 7 biomimetics-10-00408-f007:**
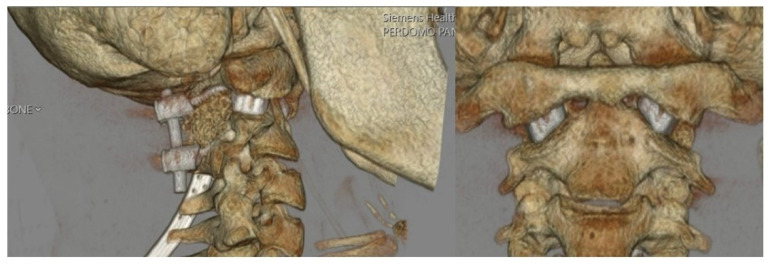
Postoperative sagittal and coronal 3D renderings of the cervical spine demonstrating implant location and surrounding bony anatomy.

**Figure 8 biomimetics-10-00408-f008:**
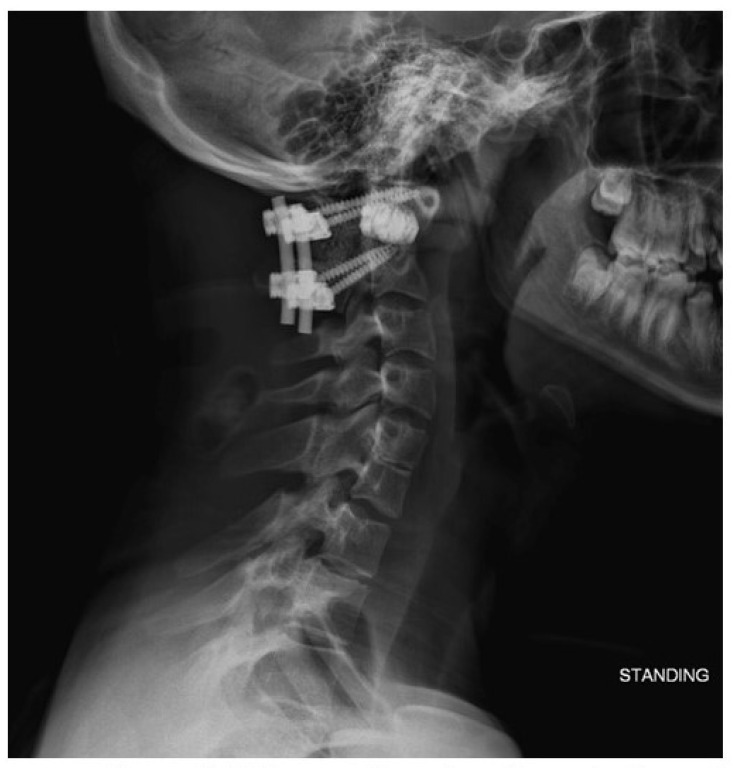
Six-week follow-up lateral radiograph demonstrating spacer placement and intact instrumentation with preserved spinal alignment.

**Figure 9 biomimetics-10-00408-f009:**
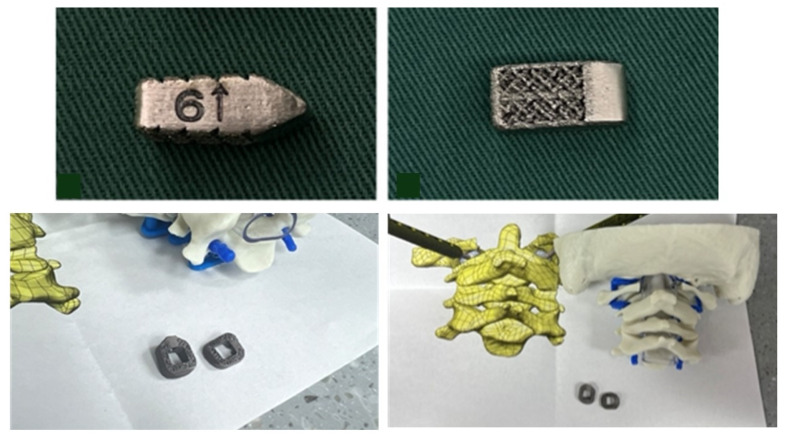
Bullet shaped custom 3DPI cage used by Jian et al. (**top**) with a relatively narrow base compared to our own custom implant (**bottom**). The bullet implant’s small footprint may increase subsidence risks. Image adapted from Jian et al. with permission per the Creative Commons CC BY license [12].

## Data Availability

The data used for this study were obtained from electronic medical records from Barnes-Jewish Hospital (St. Louis, MO) and contain protected health information. Due to institutional policies and patient confidentiality regulations, this data is not publicly available. Data access is restricted to approved researchers in compliance with ethical and regulatory requirements. Please contact Washington University’s Institutional Review Board for further information on data access.

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
