# Peer review of "Using Patient-Specific 3D-Printed C1–C2 Interfacet Spacers for the Treatment of Type 1 Basilar Invagination: A Clinical Case Report"

_biomimetics, 2025, doi:10.3390/biomimetics10060408_

Round 1
Reviewer 1 Report
Comments and Suggestions for Authors
Dear Authors!
The paper “Patient-Specific 3D-Printed C1-C2 Interfacet Spacers for the 2 Treatment of Type 1 Basilar Invagination” aims to offer a new approach for the treatment of a disease traditionally treated with distraction and stabilization through fusion of the C1-C2 vertebrae. I found this publication be a valuable and very good guideline for further surgical improvement. Even though based on the article name I expected that there will be more information about 3D printing and the technology, but it is all about the surgical method so I suggest the authors to be more specific regarding the name of the article.
- Introduction
Can you add a reference in Line 58?
Can you describe in a few sentence the used 3D printing method, the advantages and disadvantages of this technique?
- Patient information and Presentation
The clinical presentation, imaging and diagnosis part is well written.
- Materials and Methods
Line 142: one dot is missing
3.1. Preoperative Planning and 3DPI Design
I am aware with the meaning of proprietary data but as the name of the article is about „3D printed spacers” and not the surgical method or surgical case report about a 3D printed spacer and I find it misleading and confusing as my research area is 3D printing not neurosurgery and in the abstract it is not mentioned that I won’t read a word about 3D printing.
What is the base of this method, are the manufactured samples characterized? How they were sterilized and how the adequate sterilization was examined? I am sure that the composition is classified but was it investigated? Did they tested the biocompatibility properties of these samples, like long-term cytotoxicity experiments or biofilm formation?
3.2. Surgical Approach
This section is well described.
- Results
The results are well described about the surgical outcome.
- Conclusions
It is well described.
- References
The amount of reference is not adequate. Only 11 reference is mentioned whom 10 is from the last five years. I suggest the authors to add at least 10 references.
The use of the references in most of the article is consistent. All citation have to be at the end of the sentence.
Author Response
Comments 1: Even though based on the article name I expected that there will be more information about 3D printing and the technology, but it is all about the surgical method so I suggest the authors to be more specific regarding the name of the article. |
Response 1: Thank you for pointing out the possible confusion from our working title. We have updated the title to “Using Patient-Specific 3D-Printed C1-C2 Interfacet Spacers for the Treatment of Type 1 Basilar Invagination: A Clinical Case Report” to emphasize the clinical nature of our manuscript (Lines 1-4). |
Comments 2: Introduction: 1. Can you add a reference in Line 58? 2. Can you describe in a few sentence the used 3D printing method, the advantages and disadvantages of this technique? Response 2: We have added a reference providing treatment algorithms to line 58 to provide further support for the claim that different BI types require different treatment strategies. Details pertaining to the 3D printing process is detailed from lines 165-182. |
Comments 3: Materials and Methods 1. Line 142: one dot is missing Response: Thank you for pointing out this typo. We have corrected it with addition of a period. 3.1. Preoperative Planning and 3DPI Design 2. I am aware with the meaning of proprietary data but as the name of the article is about „3D printed spacers” and not the surgical method or surgical case report about a 3D printed spacer and I find it misleading and confusing as my research area is 3D printing not neurosurgery and in the abstract it is not mentioned that I won’t read a word about 3D printing. What is the base of this method, are the manufactured samples characterized? How they were sterilized and how the adequate sterilization was examined? I am sure that the composition is classified but was it investigated? Did they tested the biocompatibility properties of these samples, like long-term cytotoxicity experiments or biofilm formation? |
Response 3: Thank you for pointing out this limitation in our methods section. We agree that providing additional detail about the manufacturing process would further enrich the manuscript and thus have added details about the material composition validation and sterilization technique as follows: “The material composition of the implant was derived from biocompatible titanium and tested per ISO 10993 to ensure biocompatibility of both the material and manufacturing process. The methods and materials were previously accepted and used with other biocompatible implants. The 3D-printed devices were steam sterilized and underwent successful validation testing for 1-time use devices per industry standard,” (lines 178-182). Comments 4: The amount of reference is not adequate. Only 11 reference is mentioned whom 10 is from the last five years. I suggest the authors to add at least 10 references. Response 4: We agree that having an inadequate number of references may weaken the claims of our study. In response, we have added >10 additional references that support our statements throughout the text. Please refer to the references for a complete list (lines 365-419). |

Reviewer 2 Report
Comments and Suggestions for Authors
line 35, the conclusion needs to be strictly associated to the follow up time.
line 58, needs citation
lines 51-65, try to use other references this big part of the introduction is the summary of a single source.
line 66 needs citation
line 75, detail the novelty in relation to the previous study, you may move this in the discussion.
line 152, high quality means nothing, detail the technical specifications of the imaging that has been taken.
line 199, be quantitative in describing the procedure including anything that has been measured.
line 249, define complicated cases
Author Response
Comments 1: line 35, the conclusion needs to be strictly associated to the follow up time.
Response 1: Thank you for pointing out a limitation of our current study. Follow-up time is restricted to early periods in this case. We agree that the conclusions of our study should be considered with appropriate timing, so we have highlighted this necessary association on lines 40-41: “Future optimization of the workflow and characterization of long-term patient outcomes should be explored for these types of 3DPI.”
Comments 2: line 58, needs citation, line 66 needs citation
Response 2: Thank you for pointing out this limitation. We have accordingly added citations for these statements (line 59, 67-68)
Comments 3: lines 51-65, try to use other references this big part of the introduction is the summary of a single source.
Response 3: We agree that diversifying the references for the introduction would strengthen scientific rigor. We have added additional references/cross references that further support the statements given in the introduction. Please see lines 56, 59, 64, 67-68, 71, 73, 75, and 78.
Comments 4: line 75, detail the novelty in relation to the previous study, you may move this in the discussion.
Response 4: We agree that emphasizing the novelty of our study is a key element of this case report, so we included a statement in the introduction that previews that limitation of the previous study and how it differs from our described case. “While a precedent has been established for use of custom 3DPI in spinal oncology or deformity surgery, we were only able to identify one prior study from China evaluating custom 3DPI for basilar invagination that featured persistent risk of subsidence.” (line 75-78). Further details about the novelty of our study is provided in the discussion (line 286-294, highlighted).
Comments 5: line 152, high quality means nothing, detail the technical specifications of the imaging that has been taken.
Response 5: Thank you for highlighting the unclear wording. We agree that “high quality” is not specific and have provided details about the CT scanner and image quality. “Photon-counting detector CT (PCD-CT) scans were performed using a Siemens NAEOTOM Alpha system (software version VB10A; Siemens Healthineers, Germany). Cervical spine (C-spine) studies were reconstructed at two slice thicknesses—3 mm and 0.4 mm— using the Br44f (medium-sharp) and Br64f (ultra-sharp) kernels, respectively. All reconstructions incorporated Quantum Iterative Reconstruction (QIR) at level 3. Additional details may be found in the supplemental data.” (lines 155-161). Additional details have also been added to the table given in the supplemental data.
Comments 6: line 199, be quantitative in describing the procedure including anything that has been measured
Response 6: We agree that quantitative metrics should be used when describing the results of the procedure. The sentence following line 199 includes quantitative descriptors as follows: “Postoperative ADI was 3 mm, MLV was 2 mm, and CXA was 134°. In other words, the ADI decreased by 0.7 mm, MLV decreased by 7mm, and CXA increased by 15°.” (lines 214-216)
Comments 7: line 249, define complicated cases
Response 7: We agree that “complicated cases” should be defined, and so we added a few examples of what could constitute as complicated and corresponding references that define such cases in which 3DPI may be useful: “Overall, for complicated cases associated with longer operative times, greater surgical invasiveness, or greater post-operative complications, 3DPI are a useful surgical option when applicable [10,16,18,19].” (lines 262-264)

Reviewer 3 Report
Comments and Suggestions for Authors
The manuscript presents a novel application of patient-specific 3D-printed interfacet spacers for Type 1 Basilar Invagination (BI). The broader base design of the spacers addresses subsidence risks and preserves cervical range of motion, offering a tailored solution for the adolescent patient.
The detailed clinical presentation, imaging analysis (CT, MRI), and surgical workflow (including preoperative planning, FDA approval, and postoperative outcomes) are thorough and well-documented.
The case highlights the potential of 3D-printed implants (3DPI) to address complex craniovertebral junction (CVJ) pathologies with dysplastic anatomy, providing a proof of concept for future applications.
As a case report, the study lacks statistical power or comparative analysis. The authors appropriately acknowledge the need for a larger series to evaluate long-term outcomes.
Reproducibility may be limited due to undisclosed proprietary methods (e.g., the PURI-TI protocol).
In summary, the manuscript is well-written and clinically significant, offering valuable insights into the use of 3DPI for BI. While the single-case nature limits broader conclusions, the detailed methodology and successful outcomes support its publication as a feasibility study that encourages further research into customised implants for CVJ pathologies.
Author Response
Thank you for taking the time to review this manuscript. We appreciate the positive feedback and hope you enjoyed reading about our experience and findings for this case report.
Round 2
Reviewer 1 Report
Comments and Suggestions for Authors
Dear Authors!
Thank you very much for correcting the requested corrections, so that the publication is of a higher quality.
I hope in the future I can read more about the used manufacturing process.
Reviewer 2 Report
Comments and Suggestions for Authors
thanks or having addressed my comments